# The Nature of Religious and Spiritual Needs in Palliative Care Patients, Carers, and Families and How They Can Be Addressed from a Specialist Spiritual Care Perspective

**Kate L. Bradford** 

Westmead Hospital, Cnr. Hawkesbury Road and Darcy Road, Westmead, NSW 2145, Australia;
kate.bradford@health.nsw.gov.au

**Abstract:** This paper is written from the perspective of a specialist religious and spiritual care practitioner who practises in major referral hospitals in Sydney, Australia. In these hospitals, religious care and ward-based spiritual care chaplaincy services are offered in tandem. The perspective offered is based on the author's knowledge, research, and experience in how people make the necessary religious or spiritual adjustments to their deep view of reality when faced with severe or chronic pain or terminal illness. Religious and/or Spiritual Care (R/SC) are interventions that scaffold people through a process of sense-making which helps them find meaning in their new reality. These R/SC adjustments concern conscious and unconscious beliefs about ultimate meaning, morality, justice, transcendence, and relationships within themselves, and with others and the supernatural. Palliative care practitioners described the importance of spiritual care and integrated spiritual care principles into their biopsychosocial-spiritual model of care in the 1960s. As palliative care practitioners have increasingly clarified their discipline as a distinct discipline in medicine, religious and spiritual care practitioners have struggled to define their place. These concerns merit a fresh evaluation of the religious and spiritual needs of patients, carers, and families of those suffering chronic pain and terminal illness and further clarification of the specialist discipline of religious and spiritual care.

**Keywords:** palliative care; religious care; spiritual care; narrow and broad specialist religious and spiritual care; sense-making; finding meaning

## 1. Introduction

This paper is written from the perspective of a specialist religious and spiritual care practitioner who has practised for many years in major referral hospitals in Sydney, Australia. Australia is becoming increasingly secular, like many Western countries, with 38.9% reporting no religious affiliation in a recent 2021 census. When presented with the option of recording no religion, over 60% of Australians still recorded a religious affiliation, with these percentages varying widely between regions. The social context which gives rise to the viewpoint of the author of this paper is the multicultural context of the Western Suburbs Local Health District (WSLHD), where Westmead Hospital (958 beds) is located, serving a population of 1.3 million people. The residents of this region of Sydney recorded far higher religious adherence than Australia's general population, with almost 80% choosing to nominate a religion, and just 20.5% nominating no religious affiliation. The population of this region identified as 44.2% Christian (of which 24% are Catholic, 5.9% are Anglican, and 3.6% are Orthodox), and 9.4% are recorded as Islamic, 10.3% Hindu, 5.8% Buddhist, 0.6 Sikh, and 0.05% Jewish (Australian Bureau of Statistics 2022). These statistics also correspond to a significantly religious patient population and healthcare workforce. The chaplaincy department of Westmead Hospital provides specific religious care from Catholic, Protestant, Islamic, Hindu, and Buddhist traditions in tandem with ward-based spiritual care services offered to all patients, families, carers, and staff. The

author hopes that this perspective will provide an adequate review of religious and spiritual care within the discipline of palliative care and provide a comprehensive way forward for religious and spiritual care practitioners as they support people as they form new or deepened understandings, ways of being, perceptions, and experiences in their present horizontal and vertical aspects of reality as expressed through their spirituality.

## 2. Spirituality, Religiousness, and Personal Beliefs

Over the last twenty-five years, there have been concerted efforts to describe and integrate the spiritual domain of human need into the overall health care of a person. Spirituality as a discipline in health care is relatively recent, with the word 'spirituality' (Kruizinga et al. 2018) not appearing in Medline until the 1980s. A working group of the World Health Organisation (WHO) extensively discussed the 'non-material' or 'spiritual' dimension of health (Kruizinga et al. 2018, p. 4). In 1998, the WHO working group proposed a new definition of health, as 'a dynamic state of complete physical, mental, spiritual and social well-being and not merely the absence of disease or infirmity (Kruizinga et al. 2018)'. Concerning health and quality of life and the way in which they interrelate, the WHO working group considered the spiritual domain to encompass the concepts of spirituality, religiousness, and personal beliefs (SRPB) (Kruizinga et al. 2018, p. 5).

This domain/facet now examines an individual's personal beliefs and how these affect quality of life. This might be by helping the person cope with difficulties in their life, giving structure to experience, ascribing meaning to spiritual and personal questions, and, more generally, providing the person with a sense of well-being. This facet addresses people with differing religious beliefs (including Buddhists, Christians, Hindus, and Muslims) and others with differing beliefs that do not fit a particular religious orientation. Religion, personal beliefs, and spirituality are sources of comfort, well-being, security, meaning, a sense of belonging, purpose, and strength for many people. However, some people feel that religion negatively influences their life (Kruizinga et al. 2018, p. 6). That religion or, indeed, any spirituality or belief has a negative influence on some people indicates that these are not an undisputed good in and of themselves but may have positive or negative valences, which influence the flourishing or languishing of the individual (Pattison 2010, p. 359).

Formal religions can be observed as having four Cs: creeds, codes, cult, and community. The creeds are stated beliefs, the codes relate to conduct, and the cult is expressed through ritual by which people of a community locate themselves in the world with reference to ordinary and extraordinary powers, meanings, and values. (Albanese 2012, pp. 9–10). Many people describe themselves as spiritual and differentiate themselves from formal religion; however, Liselotte Frisk observed that those who describe themselves as spiritual rather than religious demonstrated marked similarities to the mystical tenets of the major religions. Frisk noted six characteristics that bore a resemblance to mystical rather than formal religion. These are (1) a shift from particular to eclectic—rather than relying on one source or one religion, world religions are viewed as resources; (2) a shift from dogma to experience; (3) a shift from collective to personal with an emphasis on healing, wellbeing, and relationality; (4) a shift from hierarchical to egalitarian religious view, where God, or Other, is not regarded as above but rather infused through everything; (5) a further shift from theological to anthropological that emphasises subjective feeling and a personal growth dimension. The final shift mentioned by Frisk is (6) a shift in focus from life after death to this-worldliness with more emphasis on a spiritual life now, rather than life after death (Frisk 2009, pp. iv–vii). Frisk's analysis indicates an underlying religiosity even in the views of 'the spiritual but not religious', demonstrating that these views have some religious aspects in common with the religious views from which they differ.

There is a further group whose personal beliefs are neither religious nor spiritual. Various combinations of religious, spiritual, and personal beliefs can be beneficially summarised and abbreviated as religious and spiritual (R+/S+), religious but not spiritual (R+/S−), spiritual but not religious (R−/S+), and those who are neither religious nor spiritual (R−/S−). A further broad differentiation can be made within the R−/S+ between

individuals moving away from formal religions and individuals who have had little or no contact with formal religion. This distinction is critical, as those who have been exposed to or are part of organised religions bear closer similarities to those with religious worldviews than those with little or no contact with religious worldviews.

The spiritual domain comprising SRPB is recognised as a cornerstone of the holistic care framework employed in palliative care, known as the *biopsychosocial-spiritual* model. This includes the physical, social, psychological, and spiritual domains. Palliative care practitioners were the first representatives of a medical discipline to describe the importance of spiritual care and to integrate spiritual care principles into their biopsychosocial-spiritual model of care in the 1960s. The origins of palliative care and the hospice movement can be traced back to nineteenth-century Catholic and Protestant religious orders which provided rest houses or hospices to care for people with terminal illnesses. Nurse, social worker, and doctor Cicely Saunders (1918–2005) saw the need for a facility that cared holistically for people with terminal illnesses, which addressed their total pain, taking into account patients' medical needs, physical pain, emotional suffering, and social and spiritual needs. Saunders herself was a Christian and was deeply influenced by the person-centred care of Swiss physician Paul Tournier (1898–1986), who was also concerned for a person's spiritual well-being. When Saunders established St Christopher's Hospice in London, she initially investigated creating a religious community. Instead, in consultation with others, she settled on providing an environment based on a Christian religious foundation that encouraged all to contribute in their own way, that 'love is the way through,' given in care, thoughtfulness, prayer, and silence, and patients must have a sense of security and support and be given the opportunity to find peace, without being subjected to special pressures. There was a chapel for prayer and worship (Clark 2016, pp. 85–97).

In some aspects, palliative care was always spiritual, and the two disciplines developed together in this environment. Still, as palliative care practitioners have increasingly clarified palliative care as a distinct discipline in medicine, yet religious and spiritual care practitioners have continued to struggle to define their place. Palliative care practitioners brought a unique lens to spiritual care from the outset, understanding that spirituality exists within dynamic extended systems of relationships. These complex relational networks and connections exist between patients, their families, personal carers, the interdisciplinary team, sociocultural and religious communities, and existential and transcendent values and beliefs. Much of the academic research on spirituality focuses on the individual. However, in reality, an individual's spirituality is embedded within a broader multivalent relational network.

The academic study of spirituality is focused on beliefs, perceptions, inward motivations, values, experiences, and expectations, and the way in which people resolve inner conflicts. These conflicts are between what is happening and what they believe ought to happen within themselves, with others, the environment/world, and God or some other higher power (Mount et al. 2007; Oxhandler and Giardina 2017; Abernethy et al. 2020). A recent meta-analysis by a Delphi panel that convened to review the available research regarding spiritual care to seriously ill patients concluded that 92.5% of the reviewed papers recommended that spiritual care be routinely incorporated into the medical care of patients with serious illness. The same panel emphasised that three major implications stood out from this study (Balboni et al. 2022). When used in the palliative care context, the recommendations are to (1) incorporate spiritual care into care for all palliative care patients; (2) incorporate spiritual care education into the training of all interdisciplinary team members caring for palliative patients; and (3) include specialty practitioners of spiritual care in the review and care of palliative patients. I will firstly provide an overview of spirituality, followed by frameworks for spiritual beliefs and current religious and spiritual care practices for palliative patients as outlined in their first two recommendations.

In the second part of this paper, I will address specialist religious and spiritual care, reviewing current opportunities and limitations. Much of the current literature in this area retains the tenets of two prevailing twentieth-century paradigms of spiri-

tual care—*clerical* and *clinical*. The clerical model is more heavily weighted to propositional knowledge, and the clinical model emphasises *experiential-expressive* knowledge (Lindbeck 1984; Gerkin 1997, pp. 106–8). The two paradigms have been hampered by a lack of cohesion, common language, and concepts, and an approach restricting any overarching structure around spirituality and spiritual care. This impasse directly limits the scope of research in spiritual care and the development of curricula for religious and spiritual care practice. I propose a way forward that can incorporate the clerical (theological) and clinical (psychological) aspects but extends both by placing them in conversation with other disciplines engaged in sense-making, finding meaning and future possibilities in adverse circumstances, and wisdom. In contrast to propositional and experiential-expressive modes of knowing, this integrative way is based on *cultural-linguistic* knowledge—commonly captured in the term 'wisdom' (ibid.).

### 3. A Framework for Spiritual, Religious, and Personal Beliefs

Spiritual, religious, and personal beliefs, while interrelated, differ from each other and provide different connections within the spiritual self, forming a complex matrix of beliefs, values, and expectations. The spiritual/religious/existential self differs from the social and psychological self as it has horizontal and vertical dimensions (Hood et al. 2009). The horizontal dimension comprises relationships within the self, and with others and social and natural environments. The vertical dimension engages 'the Other' and can include something beyond or 'other-than' that which can be understood by reason or rationality. The language of philosopher Charles Taylor refers to the here-and-now or horizontal dimension as the 'immanent frame' and the vertical as the 'transcendent frame,' (Taylor 2009), which includes spiritual forces, belief in God, prayer, karma, blessing or curses, and sacred or divine elements to life. Spirituality always contains horizontal elements and often has vertical elements, and religion has primarily vertical elements entwined with horizontal elements. Therefore, an important part of any religious/spiritual/existential care practice is to discern how these horizontal and vertical elements of a person's belief systems intersect with a person's spiritual self.

In this paper, I will refer to religious, spiritual, existential, and personal beliefs by the abbreviation R/S to capture both the vertical and horizontal aspects of the domain described by the WHO as 'spiritual', to mitigate against either collapsing the vertical down into the horizontal or, conversely, the equal and opposite mistake of assuming all horizontal needs and concerns are vertical needs, sometimes referred to as 'over-spiritualising events'.

#### 3.1. Religion

Historically, in the West, spiritual concerns were largely attended to through religion, as they still are in the majority of the world. From this perspective, spirituality represented the more subjective side of religious faith. However, the terms 'religion' and 'spirituality' have always had and continue to have related and distinct meanings where religions hold to observable and objective external forms. Religions are communities of practice with a common language, shared wisdom, community support, historical roots, and institutional backing (Tacey 2022). Religious language crosses the boundary of rationality and functions quite specifically to speak reality into being through prayer, chants, liturgy, blessing, and even denouncements and curses. Religious wisdom is contained in pithy sayings, proverbs, paradoxical stories, poetry, parables and the like, leaving the reader, or hearer, to puzzle and ponder about the nature of reality. From a psychological and sociological perspective,[1] ritualistic and symbolic religious practices can be observed and described.

#### 3.2. Spirituality

Spirituality is concerned with spiritual life. Similarly, Simon Peng Keller has described 'Spirituality' as a travelling term (Peng-Keller 2019), and John Swinton, notes that it has migrated from the religious to the 'secular', leading to an evolution in its meaning (Swinton 2001, p. 11). In the migration from religious to secular, some people moved to-

wards atheism, but across many populations, more increasingly self-identify as spiritual. A working definition of spirituality which includes formal religion is 'Spirituality is a dynamic and intrinsic aspect of humanity through which persons seek ultimate meaning, purpose, and transcendence and experience relationships to self, family, others, community, society, nature, and the significant or sacred. Spirituality is expressed through beliefs, values, traditions, and practices (Balboni et al. 2022, p. 186)'. Such a definition provides a practical reference point that builds consensus for a shared project while allowing for considerable diversity within the consensus.

### 3.3. Personal Beliefs and Values

It is important, however, to appreciate that not all people consider themselves as either religious or spiritual (R−/S−) and the inclusion of personal beliefs clarifies that the R/S dimension includes people who hold no beliefs in nonmaterial structures or higher powers and believe that there is nothing beyond the material world, which in turn shapes their spiritual self within the immanent frame along the horizontal dimension (World Health Organization 1998). Personal beliefs focus on a person's moral and ethical beliefs and their personal code of conduct.

## 4. The Spiritual Self

The religious and spiritual beliefs of each person form a constellation that is closely related to their core identity, comprising their spiritual self (Brandt 2019). The beliefs and values of the spiritual self are held in a web of connections and relationships encompassing a person's relationship within themselves, with others (people, pets), the world (social and natural environment) and/or a belief in a higher power or being or things beyond the physical world. When functioning well, this constellation or web of relationships provides a way of making sense of events and finding meaning. When relationships in this web come under pressure, and are challenged by new events or circumstances, the relational threads become strained and may even break while trying to bear the weight of the new reality. In these situations, a person's ability to make sense and find meaning becomes overwhelmed, and it is experienced as spiritual, religious, or existential distress. It is these signs and symptoms of distress, struggles, denial, or questioning that draw attention to the presence of R/S concerns or unmet needs.

### 4.1. Religious and Spiritual Wellbeing

R/S well-being is a state in which a person's spiritual constellation or web of horizontal and vertical relationships and connections is being maintained and restored in such a way that the spiritual self feels settled or stable. This includes how they feel in themselves, with others, their world—comprising their social and physical environment—and in matters of faith and belief, including higher powers. In the ordinary course of life, people are able to make adjustments to this web of relationships to respond to changing circumstances when there is not too much change or when the changes are not too rapid. It will also be apparent that people who struggle in these relationships have a more fragile spiritual constellation and further setbacks have the potential to put a far greater hole or tear in the person's existing web of relationships.

### 4.2. Religious and Spiritual Needs

R/S needs are very often normal responses to abnormal circumstances, and in and of themselves are not pathological; for example, extreme grief is a normal, not pathological, response to the loss of a loved one. Other such responses include feeling gutted at the betrayal of a best friend, or feeling utterly devastated at the news of a terminal cancer diagnosis. However, not all experiences that precipitate R/S needs are sudden in nature; some are made up of a continuous shortfall of R/S resources across a lifetime, and others accumulate slowly through a fraying away of the R/S resources at one's disposal, as might happen with chronic illnesses, or care for a family member with a disability or mental

health challenges. These experiences tear at the very fabric of the web of relationships that maintain the spiritual self.

The signs and symptoms of R/S needs present as concerns, distress, denial, psychological pain, needing to make a difficult decision, loss, grief, and struggles (Balboni et al. 2022). The distress can manifest in existential anguish or questioning, such as Why me? Why now? What have I done to deserve this? The R/S needs may relate to a single dimension of spiritual need, or, more commonly, the intersections of different R/S needs across various relationships within a person's spiritual self with themselves, others, their social or physical environment, and their faith and beliefs. Similarly, the process of meeting these needs will also involve these different responses with different people.

*4.3. Religious and Spiritual Care*

There are two primary modes of religious and spiritual care which correspond to the explicit nature of formal religions and the more implicit character of other spiritualities. These modes are characterised by narrow and broad approaches: (1) The narrow mode describes religious care which is provided by an authorised religious representative such as a pastor, imam, rabbi, priest, minister, cleric, or similar whose primary lens through which the care is offered is through a religious framework. The narrow religious framework is in contrast to (2) a broad spiritual care approach. The distinction is a critical one as some, but not all, specialist religious care practitioners are also trained in broad spiritual care. Practical theologian Stephen Pattison notes that the movement in health care to provide generic spiritual care neither replaces the role of substantive religious traditions nor particular individuals' need for such, and he also asks if religious specialists are the best people to provide generic spiritual care, although they bring the depth of wisdom of their religious tradition into their specialist spiritual care practice when more broadly trained (Pattison 2001, 2010). Writing also from a religious perspective, Eckhard Frick (2017) notes that 'Spiritual Care is not a synonym for [religious] chaplaincy, pastoral care etc. Neither can it be delegated as such to chaplains.' Frick is not opposed to spiritual care but wished to emphasise that there are essential differences between religious and spiritual care.

Frick observes that spiritual care 'works in a symphony of different spiritualities,' working as it does with institutional religious frameworks and eclectic spiritual searching. Additionally, he observes that individual patients and carers are also located with specific spiritualities. Easy or shallow harmonies may miss hidden depths of spirituality in people's lives (Frick 2017). R/SC providers work with the symphony of different spiritualities, ranging from broad down to narrow, with the hope of meeting people's spiritual needs while avoiding easy or shallow harmonisations. Some religious care specialists work specifically within one tradition; others are trained and qualified in both specific religious traditions and the provision of specialised broader aspects of spiritual care.

While precise definitions vary between countries and public and religious facilities, generally, in contrast to more narrow religious care, spiritual care providers (SCPs) are trained broadly to provide care for people from many faiths and non-faith traditions and philosophical backgrounds and disciplines. The focus of SC is on meaning and purpose, and on connectedness, with an emphasis on the integration of body, mind, and spirit; well-being; and connection to self, others, and/or a higher power (Handzo and Puchalski 2015, p. 171). Spiritual care providers do not necessarily represent or come from any particular religious group. In contrast to SCPs, chaplains traditionally have come from the religious sphere, and in many countries, the term 'chaplain' refers narrowly to an SCP who belongs to a specific religious tradition able to conduct religious services and rituals.

## 5. Attending to R/S Needs through Palliative Care Practice

Spiritually informed care takes account of the biopsychosocial-spiritual needs of the whole person. R/SC both cares for the spirit and is spiritual in nature. Because R/S needs are across a person's relationship with themselves, others, their social and physical environment, and support for their belief and/or faith, there are several ways in which the program

design and physical environment can facilitate spiritual well-being, including the physical environment. (Ulrich 2006) The environment includes the architectural, visual, and auditory aesthetics of the space, underscoring the importance of chapel spaces, places for quiet reflection, and windows with pleasant outlooks (Pallasmaa 2015). Attention to these details helps to create spaces where people can make spiritual adjustments to new, unsought, and often unpleasant realities. Human illness, ageing, and frailty are universal experiences for which many people seek solace as they make adjustments. Some people seek solace in sacred spaces and practices such as prayer and contemplation, and a 'well-designed health care chapel serves as a sacred meeting place where this conflict of life and illness can be encountered safely within the healthcare facility' (Lawson and Alfaro 2022, p. 93). Architecture and design not only affect physical space but also social and relational space, and programs designed to provide continuity of care by multidisciplinary teams support the dignity and worth of the patients. The architectural concept of a low-entry threshold is important physically as well as metaphorically, describing a social dynamic that has the effect of providing and creating a sense of warmth and welcome. At the level of individual care, compassionate care forms the essence of spiritually informed care.

Christina Puchalski describes compassionate care as compassionate presence (Puchalski 2008). Lasair and Sinclair regard compassion as the 'quintessential indicator and a universal medium for addressing patient and family members' spiritual needs' (Lasair and Sinclair 2018, p. 292). Furthermore, they report that 'according to patients, compassion is a multidimensional care construct, distinguishable and preferred from empathy and sympathy' (Lasair and Sinclair 2018, p. 293). Lasair and Sinclair observe that compassionate care begins simply by seeing patients and families in the 'uniqueness of their experience' separate from their medical concerns (Lasair and Sinclair 2018, p. 293). Because the healthcare staff have already formed a compassionate relationship with the patients and families, care of 'spiritual concerns can occur within routine medical care and by health care providers going beyond routine care, by taking an active interest in patients as people' (Lasair and Sinclair 2018, p. 293). In addition to compassionate presence, general spiritual care is enhanced when staff are cross-culturally competent and practice trauma-informed care (Evans and Coccoma 2014), whereby the carer is not only a safe and trustworthy person but engenders trust in the other.

*5.1. Generalist/Specialist Religious and Spiritual Care in Palliative Care*

Palliative care, in essence, seeks to provide compassionate person-centred care in the context of serious illness. As part of holistic care, patients are screened routinely for any presenting spiritual needs. Spiritual screening is performed through a simple screening tool that can be administered by any member of an interdisciplinary team. At this general level, palliative care also integrates spiritual care as part of the biopsychosocial-spiritual model of care, where all members of the team provide compassionate presence, empathic listening, and respect for the patient's religious and spiritual values through holistic, person-centred care and compassionate presence within professional competencies. To help achieve these objectives, every member of the team is encouraged to connect with their own sources of strength, hope, healing, meaning, and purpose to be able to care deeply out of their own spiritual resources. In this model of care, specialist R/SC is provided by an R/S practitioner who is part of the multidisciplinary team, and who is very often a chaplain (in the US, a board-certified chaplain and in other countries, a suitably qualified chaplain or spiritual care practitioner).

5.1.1. Spiritual Screening

A spiritual screen captures a snapshot of R/S needs across this web of relational networks, with questions regarding oneself and one's relationships and communities, and questions about religion, faith, and belief at the end of the screen, rather than the beginning, so as to indicate genuine concern for the whole person. Several spiritual well-being scales, such as the JAREL Spiritual Well-Being Scale (JSWBS) (Hungelmann et al. 1996;



Best et al. 2020, p. 5) have been developed that aim to capture both horizontal and vertical aspects of a person's spirituality by screening for the existential (meaning and purpose), the quality of relationships, and religious (faith and belief) matters. In 2009, Büssing et al. developed a Spiritual Needs Questionnaire (SpNQ) to cover a wide spectrum of spiritual needs of relevance to religious and nonreligious people, which can be administered by any staff member (Büssing et al. 2009). The SpNQ is a comprehensive instrument surveying social, emotional, existential, and religious needs, capturing the motifs of Connection, Peace, Meaning/Purpose, and Transcendence through four core dimensions: Religious Orientation (i.e., faith, praying, trust in God), Search for Insight/Wisdom (which is an existential issue, i.e., insight and truth, beauty and goodness, the search for existential answers), Conscious Interactions (i.e., conscious interactions with others, self, and environment, compassion and generosity), and Transcendence Conviction (i.e., belief in the existence of higher beings, rebirth of person/soul) (Büssing 2021b, p. 3734). The questionnaire provides an opportunity to begin communication about a patient's specific needs.

Recent research on the Scottish Patient Recorded Outcome Measure for Spiritual Care (the Scottish PROM) indicates that it may be one of the simplest and most accurate tools for identifying patients experiencing spiritual distress. The brief questionnaire asks the patients how they describe themselves from a selection of terms: religious, spiritual, both, or neither. There are then five questions about how the patients have felt over the last two weeks, which they rate as 'none of the time, rarely, some of the time, often, or all the time'. The five questions relate to the extent to which they have been able to be honest with themselves about what they were really feeling, anxiety, their outlook, sense of control, and sense of peace. The highest possible score is 20, and the lowest is zero. Patients who record a score of nine or lower are referred to a spiritual care practitioner (Snowden et al. 2018, 2022).

### 5.1.2. Spiritual History

In addition to the initial spiritual screening, it is recommended that the treating physician take a spiritual history as part of the patient's history. Harold Koenig stressed that a physician's spiritual history tool needs to be brief and easy to remember, gather appropriate information, be patient-centred, and be validated as credible (Koenig 2002). Two such well-validated tools are FICA (Puchalski and Romer 2000), and HOPE (Anandarajah and Hight 2001). Christina Puchalski explains that when a clinician attends to a patient's spiritual history, they listen deeply to what is important for the patient. They provide space for the patients to express hopelessness and despair and try to understand their suffering and hopes. Hearing a patient's spiritual history helps the clinician identify spiritual distress through the connection of compassionate presence and reflective listening, and refer to spiritual care professionals for further assessment and intervention and understand where the patient's spirituality fits within their overall care plan (Puchalski 2021, p. 35). A skilled and compassionate clinician is able to enquire about their patients, whether they are struggling with a loss of meaning or joy in their lives or they are asking difficult questions or experiencing religious or spiritual struggles (Johnston Taylor 2020).

It can be observed from the above sections that general or initial religious and spiritual care is attended to initially by a range of clinicians on the multidisciplinary team, firstly through an initial R/S screening instrument and then with a spiritual history taken together as part of the patient's medical and biopsychosocial history. There is obvious value to having staff who feel comfortable talking with patients and asking questions along the lines of how a patient is feeling in themselves to know if the patient has supportive relationships with others or not, and to enquire as to whether they are connected to their community. How does the patient experience their environment when they are well—work, hobbies, special interests, nature, the land, etc.? And in the same vein of general enquiry, it is important to know if religion or spirituality is something that is important to them or whether they would describe themselves as religious, spiritual, both, or neither. It has been noted in some studies that staff fear entering into a patient's spiritual distress without having the time or skills to address the very real struggle being expressed, and know

that some people may be likely to carry a higher load of spiritual distress and struggle than others.

There are many people in a given community who are carrying a high load of R/S needs before suffering a major crisis or life-limiting illness; indeed, these higher spiritual needs correlate directly with poorer health status both in morbidity and mortality (Hughes et al. 2017). It is important to have screening to identify these patients for R/SC assessment and support as these factors indicate the possibility of greater R/S need. Patients who suffer from intellectual disability, physical disability, or chronic health issues, poor mental health, or conditions such as schizophrenia, bipolar, and depression may well need more support adjusting to a serious life-limiting diagnosis. The second cluster of R/S need is around attachment and attention disorders, including family systems 'cut-off', abuse and neglect, and those with an autism spectrum disorder and/or attention deficit disorders. A third cluster is around the trauma of dislocation, including migration and status as refugees of climate disaster, war, and genocide. Lastly, there is a fourth cluster around social challenges which include drug and alcohol use disorders, imprisonment, violence, abuse and neglect, and people who are struggling religiously or spiritually. Pargament and Exline have detailed six areas of spiritual struggle (Pargament and Exline 2021). A further group that very often presents with high spiritual needs is that of those who have received a serious medical diagnosis with a severe life-limiting prognosis; this news affects their entire spiritual web of relationships simultaneously. People describe this experience as being shaken to the core, having their world ripped apart, or having the rug pulled out from under them. In these situations, illness and spiritual suffering may intensify each other.

### 5.1.3. Interdisciplinary Spiritual Assessments

The biopsychosocial-spiritual model bears witness to the intertwined nature of different domains of health and suffering and the importance of attempting to explore the multivalent way in which the patient believes, experiences, perceives, and relates to horizontal and vertical aspects of reality. The model also accentuates the importance of these domains and the R/SCP being in connection with the interdisciplinary team and communicating through the patient's electronic records. It is clear, however, from the number of R/S assessment instruments developed and discussions of R/S needs in such diverse areas as nursing (Highfield and Cason 1983), medicine (Borneman et al. 2010), social work (Hodge 2015), and psychology (Rego and Nunes 2019; Vieten et al. 2013), that there is a need to integrate R/SC training into the training programs of these disciplines. Vieten et al. (2013) offer an excellent list of attitudes, knowledge, and skills for conducting spiritual assessments. In a country such as Australia, where there is a diverse multicultural population and secular beliefs are held alongside strong religious and spiritual beliefs, it is best to begin with implicit assessments rather than explicit. There are at least two reasons for beginning with implicit assessments; firstly, for almost all people, family is their first spiritual support system, and secondly, even when religious beliefs are identified, this does not tell the assessor if the beliefs are held extrinsically or intrinsically. Even for religious people, creed alone is not enough; it will depend upon the way the person relates to the code, cult, and community of the religion, and the positive or negative valence of their religious beliefs.

Beginning with an implicit assessment allows for trust to develop between the patient and practitioner. Spiritual discussions depend upon a high degree of trust and authenticity to enable a person to be able to speak about what lies beneath their stated beliefs, emotions, griefs, and inner motivations. The practitioner needs to be able to work out of their spiritual self, maintaining a critical distance (without oversharing or overidentifying) in order to speak with another's spiritual self. As the assessment moves to spiritual and religious topics, the patient needs to feel safe enough to indicate the role that religion or spirituality plays in their life. These are the foundations upon which people form their opinions and make decisions. A practitioner might be wondering what there is to work with—particularly, can these beliefs bear weight or does the person want them to bear

weight? Does this belief lighten their load or add to it? What might need to change or alter to allow for further exploration of this topic? The skills for such discussion are found in various reflective practices such as spiritual direction, nondirective counselling, and soul companioning. An excellent discussion and application of this topic is *Working with spiritual struggles in psychotherapy* (Pargament and Exline 2021). Hodge suggests several assessment tools for social work and mental health professionals that facilitate spiritual conversations, including life maps, genograms, and ecograms (Hodge 2015).

*5.2. Boundaries and Barriers to Generalist R/SC*

The obvious strengths of this generalist/specialist system are that the whole service is informed by the patient's spiritual needs and all staff are offering general spiritual care through a genuine attitude of heartfelt compassion for their patients, with a desire to know how the patients are really coping within themselves and to support them in ways that are important to them personally and spiritually. When discussing parameters guiding a health care practitioner's relationship with their patients, Koenig cites Edmund Pellegrino, who offers four touchstones of wisdom: profession, patient, compassion, and consent. Profession speaks to competency, skills, and ethics; patient acknowledges that the person seeking help is suffering and vulnerable; compassion is sharing in the patient's suffering (or, as already discussed, being a compassionate presence); and consent speaks to freely given informed consent by both parties and the absence of pressure or force by either side (Koenig 2013, p. 91).

Health professionals without specialist R/SC training are limited by their expertise and usually lack the training to resolve complex spiritual problems. There is a difference between taking a spiritual history or assessment in a respectful manner and learning about a patient's religious beliefs, resources, and struggles and adequately addressing the spiritual struggles that may arise (Koenig 2013, p. 94). Health professionals can support a patient in their beliefs but are untrained in advising on spiritual beliefs, and, furthermore, need to be aware of the power differential contained in any spiritual advice given. Spiritual screening, taking a spiritual history and supporting patients in their spiritual walk, and offering a simple prayer if requested by the patients do not cross boundaries. To initiate prayer or rituals or to offer unsolicited spiritual advice is to move outside the areas defined by the prescribed boundary touchstones. A particularly grey area is when a health professional shares a religious or faith tradition with a patient which can be a great comfort to the patient, but there are risks of losing their differentiation and becoming enmeshed with the patient. In addition, some practitioners are tempted to resolve the patient's struggles by offering seemingly simple advice which misses the truth of the complexity or unwittingly imposes a religious perspective on the other.

Koenig lists, however, a number of barriers which prevent health care professionals from providing spiritual care to the best of their ability, each of which could be addressed in generalist spiritual care training:

> These include lack of knowledge, lack of training, lack of time, concerns about projecting beliefs onto patients, uncertainty on how to address spiritual issues raised by patients, belief that knowledge about religion is not relevant to medical care, belief that discussing spiritual issues is not part of the job description, and especially, personal discomfort. (Koenig 2013, p. 102)

Further obstacles to providing general R/SC were identified as the lack of a shared R/S language and concern as to how best to handle a patient's spiritual distress or miscommunication. A critical challenge was identified when the spiritual conversation was complicated by patients' overlapping physical and existential needs. There was also the barrier of the staff needing to move their immediate focus between the patients' physical well-being and spiritual well-being (Chahrour et al. 2021). Koenig's list also included lack of settledness with one's own spirituality or discomfort with the subject (Koenig 2013, p. 104). Several studies have shown that the success of the generalist side of R/SC depends upon the

degree to which staff feel comfortable initiating conversations about R/S. Self-awareness is important, as is the ability to self-reflect on one's own spirituality and mortality.

In addition to the general and religious care that are provided by all staff on the palliative care team, there is a further layer to identifying and meeting a patient's spiritual and religious needs, usually called a 'spiritual care assessment'. If, during the spiritual screen and spiritual history, it has become clear that the patient has particular religious beliefs or denominational affiliations, it is important to ask if they would like to be connected to a religious care practitioner whose practice is more narrowly aligned with the patient's religious affiliations. The situation of a patient who has deep religious beliefs draws attention to differing types of specialist spiritual care, which is further divided into two categories: narrow and broad spiritual care, the difference between which will be clarified below.

### 5.3. Narrow and Broad Spiritual Care

When a patient is well-matched with the R/SC care they need, there are obvious benefits; however, there are a large number of studies that continue to indicate that people do not feel their religious needs were adequately addressed through serious illness, either through their medical service or through their religious communities. For adherents of particular religions, religious concepts and language may lie quite close to the surface, and their narration might be told using these concepts. There is much evidence that many patients of sincere religious faith are greatly aided by their creeds, codes, cult, and community, and when matched with a religious representative that can help them access the growing edge of their religious faith and beliefs, religious coping can be observed. (Abu-Raiya and Pargament 2015). Even when an SCP has a basic understanding of the other's faith, it is possible to work with it in a general way to help a patient access their faith in a manner that facilitates coping. Yet when the depth of the patient's metaphysical problems and beliefs exceeds the competence of the practitioner, it may be necessary to refer the patient to a professional SCP or representative of the patient's own faith (Lasair 2020, p. 1536).

A further comment is needed around the provision of specific religious care, related to a trend whereby practitioners of one faith or tradition have been trained to perform religious ceremonies and prayers in another tradition (Handzo and Puchalski 2015, p. 175). This practice, however, in the first instance, seems to be a violation of the principle of congruence in patient-centred care whereby the practitioner works out of their genuine self, and secondly, this does not take seriously the extent to which religious traditions believe that words, rituals, and ceremonies speak new realities into being. This is not to say that religious care practitioners should not do all in their power to facilitate the R/S needs of others.

A detailed R/S assessment is an in-depth conversation that seeks to understand some of the complexity of the patient's presenting R/S story, conducted by a specialist religious or spiritual care practitioner, and it is to this topic that the remainder of this paper will turn.

### Specialist Religious/Spiritual Assessment

Specialist R/SC is provided within palliative care by spiritual care professionals, and chaplains provide professional assessment, counselling, support, and rituals in the matters of a person's beliefs, traditions, values, and practices, enabling the person to access their own spiritual resources (Spiritual Health Association 2020). The recent 'Spirituality in Serious Illness and Health' report highlighted the need to involve R/SC professionals in the care of patients with serious illnesses (Balboni et al. 2022). The report noted that such professionals are trained to address spiritual needs in a manner sensitive to the patient's particular spirituality—ranging from 'spiritual, not religious' to myriad religious traditions. R/SC professionals can administer more in-depth spiritual assessments and interventions and also serve as a liaison, as necessary, to patients' spiritual communities.

A patient's R/S beliefs will shape their attitudes, perceptions, concerns, values, and interactions with themselves, others, their environment, and supernatural beliefs. R/S

beliefs are part of a person's cultural identity and are influenced by a family of origin, home, and significant life events and may align with or differ from the patient's current beliefs. Some religious and spiritual beliefs are held intrinsically and form the basis of other beliefs, attitudes, and behaviours. Some beliefs are held extrinsically and operate parallel to a deeper belief system. Other beliefs are held implicitly and can only be understood through narrative as they are not always easily explained. Extrinsic beliefs operate with full observance of rituals, but again, observance does not always account for the weighting of a person's inward beliefs (Gale 2022).

Arndt Büssing describes spirituality as a complex and multilayered construct (Büssing 2021b), and specialist R/SC practitioners work across a highly complex matrix of possibilities. There is complexity in the service they provide as well as the R/S need of the patients. Büssing identifies spirituality as having different layers with *Faith/Experience* at the core, with related *Attitudes* with subsequent *Behaviours*. Büssing observes

> [that] the experiential core aspect of spirituality is difficult to access (best in narratives), while the secondary indicators (i.e., religious trust, belief in a helping God, feelings of awe, compassion, altruism and charity, prayer and meditation) can be more easily accessed and measured with standardised instruments. Several of these indicators are not exclusively for religious people but can also be found in nonreligious persons. These indicators are *not* 'spirituality' but they may be related to its distinct aspects and layers. (Büssing 2021a, p. 2)

The spiritual distress, struggles, denial, or questioning that a spiritual screen may elicit are not the core of a person's spirituality but rather signs and symptoms that may indicate tensions within their spiritual core or self. The signs and symptoms are boundary concepts regarding medicine and health care because they move from that which can be quantified or falsified into areas of belief, values, and faith, and altogether form a different knowledge base or domain of expertise. A specialist R/SCP is experienced in speaking with people about matters of life and faith.

As part of the assessment, working from the presenting, present situation, the R/SCP needs a framework for discerning the possible options that are available to the patients. Such frameworks are useful at the spiritual screen, history-taking, and spiritual assessment levels. An instrument such as the Cynefin sense-making framework (Mark and Snowden 2006; Gray 2017) can assist by sorting religious and spiritual care needs into the categories of clear, complicated, complex, and chaotic, each of which requires a distinct response:

- Clear (straightforward as cause and effect are clear)—sense -> categorise -> respond;
- Complicated (requires expertise, but cause and effect are related)—sense -> analyse -> respond;
- Complex (co-creative, working with patterns and correlations)—probe -> sense -> respond;
- Chaotic (first aid as there are no discernible patterns)—act -> sense -> respond. (Mark and Snowden 2006)

An example of a clear response to R/S enquiry may be one where the staff or patients know what they need and simply request it, such as an enquiry regarding the use of the quiet room, a service time, or a request for a Bible, prayer beads, or similar. Requests such as these can usually be resolved at the ward level by contacting the chaplaincy or spiritual care department. A complicated need may be a request for Holy Communion, end-of-life prayers, or rituals where the appropriate response is to refer the request to the person with the precise expertise to conduct the service or rituals; these requests are directed to the chaplaincy or spiritual care department for them to make the necessary phone calls and set up the visit. Such requests are highly specific and can only be met by a representative of a particular faith. Complex R/S needs fall within the complex and multilayered constructs described by Büssing and should be referred to the specialist R/SCP for further assessment. Chaotic cases, as described in the Cynefin framework, are usually emergencies, requiring complex interventions by many interdisciplinary professionals to arrest the descent into further chaos; once stabilised, R/SC will fall into one of the three categories of clear, complicated, or complex. Spirituality includes a person's wider relational network; because

of this, it includes carers and family, who may be experiencing even greater complicated, complex, and chaotic needs. Furthermore, as each person experiences events from differing perspectives and timelines, it is possible across a relational network to have complicated, complex, and chaotic needs presenting simultaneously, in which case it is important for R/SC practitioners to notice whose needs are coming into focus and being addressed and whose ongoing needs recede into the distance and remain unaddressed.

## 6. Specialist Religious/Spiritual Care

Once the type of religious or spiritual need has been identified, the R/SCP will help scaffold people through a process of sense-making which assists them in finding meaning in their new reality. These adjustments happen at a spiritual or religious level and are concerned with conscious and unconscious beliefs about ultimate meaning, morality, justice, transcendence, and relationships within themselves and with others and the supernatural.

A key role for the R/SC professionals is assisting the other with narrative integration on both the horizontal as well as the vertical domain; one approach is to focus on a person's response to contingency, a philosophical concept which describes an event which is possible but not necessary, often eliciting the responses of 'why me, or why now?' (Scherer-Rath et al. 2012; Wuchterl 2019; Kruizinga et al. 2018).

There is considerable complexity around the provision of effective spiritual care (Kruizinga et al. 2018), and expertise bridges two domains; one area is related to the many ways in which people might make sense of events and find meaning in challenging times through difficult events, and the other is the more specific aspects of particular religious belief, practice, and rituals. In common with both domains is that the practitioner is working with the patient to help understand the patient's experience and assisting the patient in reconciling between their hope and dreams and the reality of what is happening now. A spiritual assessment will elucidate some of the finer details and provide the care team with more information suggesting possible ways forward for spiritual care.

### 6.1. Competencies for Specialist R/S Care Providers

As these R/S care providers are providing scaffolding for vulnerable people's sense-making as they find new meaning in challenging circumstances, it is essential that R/S practitioners are suitably trained and qualified to be working with another person's sense- and meaning-making 'structures'. Each of these R/SCPs would be building upon their basic skills, including trauma-informed care, cross-cultural competencies, and skills in deep listening with an understanding of mental health first aid and loss and grief. Furthermore, a suitably qualified R/SCP is a person who can put another person at ease, allow them to speak of what is really on their mind, and facilitate a dynamic two-way communication that assumes that the R/SCP is a good listener and values them and their R/S beliefs, leading the patient to feel that they have been understood (Snowden et al. 2018). While patients express a preference for R/SCPs of their own faith, they also have expectations that these minimum requirements be met (Advocat et al. 2021).

### 6.2. Current Limits of Religious and Spiritual Care Interventions

Specialist R/SC has been hampered by the inability of religious and spiritual care providers to give a clear account of the interventions that they bring to their role, and further by reductionistic descriptions of care described variously as non-proselytising, a listening ear, presence, non-directive counselling (Rogers 1951), and reflective listening. This surely represents minimum competencies rather than the extent of the art of religious and spiritual care. In one report, chaplains are described as using 'many interventions, such as empathic listening, religious rituals, and prayer,' although the same report states that the chaplain's role within palliative care includes requisite training and skill sets (Steinhauser et al. 2017). Another study found that the most frequent chaplaincy interventions were prayers, blessings, faith affirmation, empathic listening, life review, and emotional support (Steinhauser et al. 2017).

It would seem that something of an impasse has been reached, and there has been a rejection of religious care as synonymous with spiritual care or what has been referred to as the 'clerical model', as it was perceived as being overly propositional or directive. The replacement by generic spiritual care has suffered from being limited to emotion-based therapies and nondirective counselling, providing only presence and space, or the clinical model, but with little focus on the possibility for the future. What has been lacking in both models—clerical and clinical—has been a thorough engagement with wisdom traditions and transformative practices. Wisdom traditions and transformative practices can be deepened to transcend the current realities by employing different lenses.

Recent literature on spiritual and religious care is beginning to focus critically and specifically on the underlying epistemological foundations of spirituality and spiritual care, including its knowledge base, methods, practices, and the validity of interventions, in addition to personal attributes. For example, Lasair notes that, on one hand, clinical pastoral education (CPE) alone may neither equip those with divinity degrees with a sufficient formation in therapeutic modalities formation (Lasair 2021, p. 15) nor fully equip them to be able to integrate their training in specific religious traditions with their professional practices in secular public institutions' multifaith environments (Lasair 2018, p. 6). On the other hand, training in therapeutic modalities and clinical pastoral education does not eliminate the need for spiritual care practitioners to be formed through a theological and philosophical wisdom tradition (Lasair 2018, p. 6).

At the very minimum, a specialist spiritual care provider or chaplain should have formal and practical training in particular wisdom, theological, religious, or philosophical schools of thought. Such a system engages with the questions of ultimate meaning, the nature of the true, the good, and the beautiful, the meaning of life and death, suffering and evil, the possibility of life after death, and morality. Secondly, a spiritual care provider or chaplain needs to be self-aware and have a capacity for self-reflection with a stable spiritual self; thirdly, they need to have skills in relational ability. In addition to supervised training, specialised R/SCPs need to be able to describe and understand the modalities that they are working with and why they choose one course of action over another.

The last part of this paper explores some growing edges for the practitioners of specialist spiritual care informed by theoretical and therapeutic frameworks to enable the practitioner to target interventions, thus facilitating the enabling of patients to draw their own conclusions about themselves and their experiences as they recount various aspects of their narratives (Lasair 2018, p. 15). The term intervention is somewhat contested, as there is a concerted attempt by the R/SCP to facilitate, empower, resource, support, or mediate rather than direct, impose, or instruct; however, I have retained the term intervention as it captures the interruptive nature of an intervention that can support a person through a change of direction or course of events. These interruptions or interventions may appear minimalist, but when strategically used, they aid a person in making sense of their reality and finding new meaning in their relationships, situation, and narrative (Lasair 2018, p. 15).

### 6.3. Competencies, Modalities, and Interventions

An overview paper on hospice and palliative care chaplaincy from the US envisions three levels of competency within specialist religious and spiritual care: (1) foundational, (2) advanced, and (3) expert, which represents a progressive enhancement of the R/SCPs' knowledge, skills, and practices leading to greater contributions to the care of the patient, the multidisciplinary team, and the organisation (Parker et al. 2021, p. 12). A growing edge for the training of R/SCP lies in identifying interventions, modalities, frameworks, and lenses that would enrich and extend R/SC practice. Having a working understanding of theories of the spiritual self and ultimate realities, and practical wisdom as to how people change, mature, transform, and transcend negative circumstances, enables the practitioner to encourage the patient toward new possibilities. Academic disciplines that to some extent overlap with religion and spirituality in terms of sense-making and finding those things that are meaningful include philosophy, psychology, sociology, linguistics, anthropology,

neurobiology, cognitive science, and narratology; the following are a selection of crucial works and researchers that could bear further research for the extending of the practice of R/SC.

### 6.3.1. Philosophical

Areas of study that have been connected with R/SC from a philosophical perspective include Charles Taylor's concepts of the immanent and transcendent frames, horizons of significance, social imaginaries, and cross-pressure (Taylor 2009); Paul Ricœur's orientation, disorientation, and reorientation, second naïveté, and hermeneutical ways of knowing (Ricœur 1995); Merleau-Ponty's phenomenology of perception and work on embodiment and how people inhabit space (Merleau-Ponty 2012); Kurt Wuchterl's religious and philosophical concepts of contingency (Wuchterl 2019); Martin Heidegger's articulation of the vertical and horizontal concept of reality (Heidegger 1958, pp. 18–26); Michael Polanyi's work on personal knowledge and the tacit dimension, as well as levels of explanation (Polanyi 1966, 1969; Polanyi and Prosch 1975); Roy Bhaskar's strata-of-reality (Bhaskar 2011); Blaise Pascal's *esprit de finesse et geometrie* (in the spirit of geometry and finesse) (Pascal 1995); and Viktor Frankl's will to meaning and logotherapy (Frankl 1969).

### 6.3.2. Psychoanalysis and Psychology

From the field of psychoanalysis, studies on theories of object relations (Winnicott 1984), attachment (Ainsworth and Bowlby 1991), unconscious and shadow self, ego and self, and the collective unconscious provide insight into the way people perceive themselves in relation to others, society, and higher beings. From the field of psychology, six key spiritual struggles have been outlined by Pargament and Exline (2021): struggles of ultimate meaning, struggles with the Divine and doubt, and the experience of struggles with moral, interpersonal, and demonic dimensions. These provide insight into the main areas a person may be struggling spiritually. Other researchers have documented the process of religious and spiritual maturity, as in James Fowler's stages of faith (Fowler 1981) and M Scott Peck's stages of spiritual development (Peck 1978). Early twentieth-century work in R/SC focused on emotions; as time went on, some understanding of the psychological and social impact of adverse childhood effects (ACEs) and family systems became critical for identifying R/S needs and distress (Bowen 1966).

### 6.3.3. Communication and Linguistics

As an R/SC's primary mode of practice is through conversation, a crucial element in any preparation must include theory and practice around connecting, communicating, linguistics, and conversations and the ways in which verbal, nonverbal, and symbolic communication work. Communication theories that impinge on R/SC include speech-acts theory, nonviolent communication (Rosenberg 2002), Socratic questioning, insights from the Johari window (Luft and Ingham 1961), and the ways in which symbols, signs, and metaphors function (Lakoff and Johnson 2003); poetry, proverbs, wisdom sayings, and the ways in which religious language (Hobbs 2020), music, and liturgy function. Much verbal communication is contained within narrative and explanation, as is seen, for instance, in the way in which Mieke Bals' text, story, and *fabula* have been used to good effect in R/SC (Bal 1990).

### 6.3.4. Neurobiology

The most recent field to enter R/SC is the burgeoning field of neurobiology. Key contributors to this overlap of disciplines are in the areas of polyvagal theory, polyvagal safety, and the window of tolerance as described by Steven Porges (2011, 2021), and the embodied effects and healing of traumatic stress as described by Basel A. Van der Kolk (2015). Ian McGilchrist's work on the hemispheres of the brain highlights the way in which focus, perception, and attention can be associated with conflict within ourselves (McGilchrist 2010), as we move between the schematized information of the left hemisphere and the holistic

knowing of the right hemisphere (McGilchrist 2010, 2021). Sarah Peyton's work with resonance is another example of a modality that overlaps with embodied R/SC (Peyton 2017).

6.3.5. Religious Practice

A range of religious and spiritual practices have been reinterpreted in the light of R/SC; perhaps the best known is the concept of presence, arising from Catholic spirituality, as with spiritual direction with its attendant disciplines of insight, support, accompaniment, direction, working with experiencing God, relationship with God, and God's leading. The ancient concept of lament arises from ancient Near Eastern literature, generally and particularly in the Hebrew Old Testament, as described by practical theologian John Swinton (2007). The religious works of Martin Buber, Paul Tillich, and Paul Tournier have shaped much of the twentieth- and twenty-first-century R/SC, including the concepts of *I-Thou* (Buber 1937), Paul Tillich's ultimate reality (Tillich 1952), and medicine of the person (Tournier 1965). More recent religiously informed work includes Religiously Informed, Relationally Skilful Chaplaincy Theory (RIRSCT) (Ragsdale and Desjardins 2022) and Confrontation (Hilsman and Walker 2022).

Mark Vernon (2022) identifies seven areas of spirituality which differ from emotional or mental health, but are more closely related to volition. The areas outlined are a self-understanding of human spirituality, what possibilities of freedom are open to people and with what should they align themselves, to know the simplicity that lies at the centre of things, resonating with the ups and downs of reality, the need to pass through death on a daily basis as well as the end of life, virtue, and understanding time as a lived reality.

When confronted by such a wide and varied list across such a range of disciplines, it can be difficult to know how an R/SCP might move beyond a situation yet avoid applying different theories and frameworks in an eclectic and piecemeal fashion. The work by philosopher and cognitive scientist John Vervaeke on meaning provides one possible overarching framework. Vervaeke speaks of different types of knowledge that could be described as cognitive, embodied, contextual, and relational; his technical terms for these types of knowledge are propositional, procedural, perspectival, and participatory, respectively as outlined in Figure 1 (Vervaeke and Ferraro 2013).

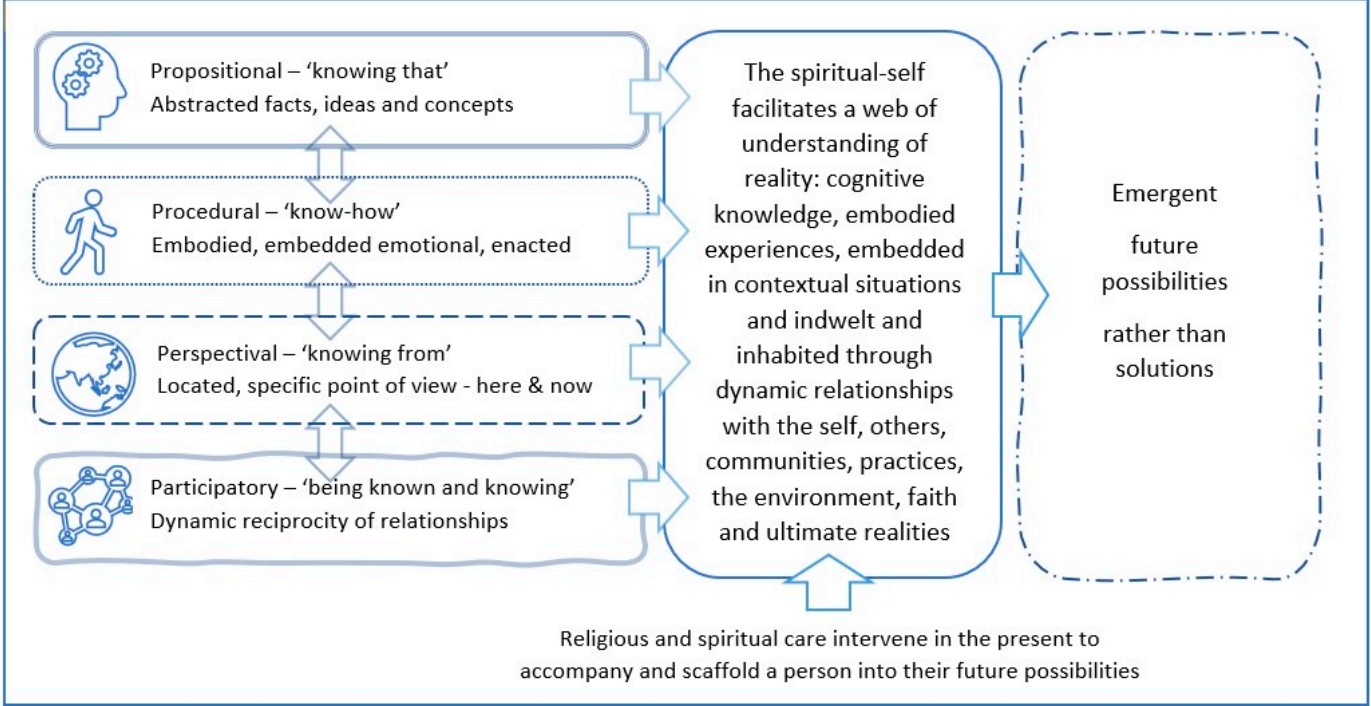

**Figure 1.** The four ways of knowing, drawing on work by Vervaeke and Ferraro (2013).

### 6.4. R/SC Framework for Scaffolding R/SC Interventions

As an R/SCP enters the complex field of an R/SC encounter, and as they listen, they need to begin to locate where the centre or focus of the patient's concerns is at that time and think about how they can connect to their world and communicate with them in both the horizontal and/or vertical dimensions.

#### 6.4.1. Propositional Knowing—Cognitive

At the propositional level are concepts, facts, ideas, and explicit beliefs. There can be great tension and cognitive conflict between what is being experienced and what is believed to be possible. Cognitive dissonance lies at the heart of many spiritual and religious struggles (Pargament and Exline 2021), particularly those around ultimate meaning and doubt. To begin to open these up, it is necessary to work in the present time and with the presenting reality. Sometimes this has been described as working with the real, not the ideal. Philosophically it has been termed working with the 'is', not the 'ought'. Wuchterl's religious and philosophical concepts of contingency exemplify the struggles and cognitive dissonance when something negative that 'was possible but not necessary' occurs and people feel the rug has been pulled from under their feet, and use such phrases as 'bad luck' or 'misfortune' or 'judgement'. Such events can trigger an existential crisis because even if one can explain the event, it does not explain why this event happened when it did to this person. Such existential questions cannot be resolved at the cognitive level; they require other forms of knowledge. The resolutions do not ultimately lie in cognitive mastery but in a different form of knowing that can accommodate paradox or mystery (Wuchterl 2019).[2]

#### 6.4.2. Procedural Knowing—Embodied

Whereas propositional knowledge is held in words, concepts, ideas, and timeless facts, procedural knowledge is not held in words but in the body; it is know-how, intuition, skill, feelings, emotions, sensations, tacit knowledge, habits, and habitation. Trauma-informed care, the polyvagal theory, and the window of tolerance all serve to inform embodied communication. Communication at the embodied level connects through emotions, experience, memory, actions, and shared space and is often beyond literal words depending upon ritual, liturgy, metaphor, poetry, and prayer. Procedural knowledge witnesses the spiritual experiences of people who may not have the ability to self-reflect or have an 'inner-life'. Many people who struggle with mental and intellectual disabilities are very awake to 'the other' and have dynamic and active religious and spiritual lives. An encounter with 'the other' while having a cognitive element is not ultimately apprehended through reason. This phenomenon has been well documented by Henri Nouwen, Oliver Sacks (1987), and paediatric oncologist Dianne M Komp (1992), and John Swinton (2016) has added considerably to this body of work from the multiple perspectives of nursing, mental health and disability care, and practical theology.

#### 6.4.3. Perspectival Knowing—Contextual

Perspectival knowing acknowledges that people understand and experience events from their own viewpoints, which can obscure insights that are held from different points of view. Philosopher Charles Taylor speaks of horizons of significance, where events are viewed through different lenses or seen from a more distant vantage point. Narrative therapy is a transformative method that uses different narrative perspectives to help people transcend difficulties through the use of narrative tools such as metaphor, externalisation, alternate outcomes, reauthoring, mapping, 're-membering', or reframing. R/SCP Simon Lasair has written extensively on the use of narrative in R/SC and has developed HAVE-H: Five Attitudes for a Narratively Grounded and Embodied Spirituality (Lasair 2021). Other effective narrative models are Reasonable Hope (Weingarten 2004, 2010); Breadth of Hope, described by paediatric oncologist Chris Feudtner (2009); and Dignity Therapy, developed by Harvey Chochinov (2012). The depths of perspectival knowing have, for centuries, been accessed through enigmatic religious sayings such as parables or Zen koans, which invite

the hearer to a space of a puzzling new perspective. Parables often begin with a familiar or sensible situation and then, through comparison with something seemingly unlikely, unsettle the hearer, drawing them into new layers of meaning or casting new light on one's own situation.

6.4.4. Participatory Knowing—Relational

The most integrative of the four types of 'knowing' is participatory, with the dynamic reciprocity of 'I-Thou' knowledge of persons (Buber 1937). The reciprocity of participation goes beyond people to the mutual indwelling of ideas, being in the world and shaped by the world, spiritual forces and beings, and attachment to objects and relations, which are further explored through dignity therapy, family systems, attachment theory, resonance theory, community, communion, prayer, and ritual. The R/SCP enters into a dynamic relationship with those they care for, sharing genuine hospitality, welcome, consent, trust, and fellowship as they mutually seek the emerging reality.

There is the potential for the R/SCP to work closely with the palliative patient while being conscious of which dimensions of reality are presenting themselves and helping the patient find what is 'real' and emerging as pain and illness are diminishing much of life around them. It takes particular skill and humility by the R/SCP to work in the middle of confusion, disappointment, distress, paradox, ambiguity, enigma, and opaque mystery and yet to facilitate the others' experience of transformation and transcendence in their suffering as they move towards a deeper religious or spiritual encounter with the 'real'.

The work of religious and spiritual care involves working with the patient to help them to access as many areas of their vertical and horizontal understandings as they are able, to continue to engage with reality, and to find meaning in the midst of suffering.

**7. Conclusions**

In agreement with the Delphi panel cited at the outset of this paper, the priorities for religious and spiritual care in palliative patients are threefold: firstly, it is crucial that R/SC is incorporated into the medical care of all palliative patients through routine spiritual screening and ensuring that treating specialists are confident to take a spiritual history; secondly, the adoption of the generalist/specialist model should be accompanied by spiritual-care education and training for all members of the interdisciplinary medical team; and thirdly, care should include speciality practitioners of religious and spiritual care for patients with serious illness, for further spiritual assessment if required. In addition to general and specialist care, within specialist care, there is a further differentiation between narrow and broad R/SC. Narrow specialisation refers to specific religious and faith traditions and authorised RCPs who can work in a particular faith or religious tradition. Specialists with broad spiritual qualifications work with people across a wide spectrum of beliefs and personal and existential situations. Related to the last point, there is a further need for new education pathways to more adequately broaden the capability frameworks for specialist religious and spiritual care practitioners.

**Funding:** This research received no external funding.

**Institutional Review Board Statement:** Not applicable.

**Informed Consent Statement:** Not applicable.

**Data Availability Statement:** Any data cited has been previously published and included in the references.

**Acknowledgments:** I wish to acknowledge with thanks Katherine Allsopp FRACP, FAChPM, Staff Specialist in Supportive and Palliative Medicine, Westmead Hospital, who kindly read this manuscript, commenting particularly from a medical and palliative care perspective. Katherine made a number of incisive observations, all of which I have included, clarifying several points and sharpening the paper; however, I take full responsibility for omissions and other points requiring further clarification.

**Conflicts of Interest:** The author declares no conflict of interest.

## Notes

[1] Contributors to this view of religion were psychologists, sociologists, and theologians exemplified by Carl Jung, Emil Durkheim, Peter Berger, Clifford Geertz, and Ernst Troeltsch.

[2] Several research projects looking at spiritual care in palliative care patients with terminal cancer have sought to empirically test notions of contingency as described by German philosopher Kurt Wuchterl.

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
