# Peer review of "The Nature of Religious and Spiritual Needs in Palliative Care Patients, Carers, and Families and How They Can Be Addressed from a Specialist Spiritual Care Perspective"

_religions, doi:10.3390/rel14010125_

Round 1

Reviewer 1 Report

This is a very interesting and well-written paper concerning religious and spiritual care in palliative care. I congratulate the Authors on their fruitful work. Apart from my overall positive opinion about the paper, I have some minor comments that can strengthen the paper:

(1) The title of the paper can be shortened and modified a little, e.g., The Nature of Religious and Spiritual Needs in Palliative Care Patients, Carers, and Families and How They Can be Addressed from a Specialist Spiritual Care Perspective.”

(2) It would be useful if the Authors mentioned religious/spiritual coping in their paper, as this construct is highly relevant to the paper’s topic.

(3)  It is not clear why the Authors provide statistics, particularly for Australia. I presume that they come from Australia. Nevertheless, the reason for giving the examples from Australia needs to be explained to provide some sociocultural context for a reader.

(4) Could the Authors expand on the topic of the similarities and differences between spirituality and religiosity?

(5) Do the Authors consider the cooperation between therapists and clergy as a means for taking care of patients’ religious/spiritual functioning? It would be useful if the Authors discussed this topic.

(6)   Could the Authors discuss common problems therapists encounter when introducing religious/spiritual topics into diagnosis and treatment and how they can be addressed? These papers may be useful: Cornish et al. (2012), Hathaway et al. (2004), and Rosmarin et al. (2013).

(7)  The Authors are encouraged to refer to the competencies for psychologists in the domains of religion and spirituality (see Vieten et al., 2016) since they can provide a reader with clear guidance concerning the required and preferred competencies in this area.

(8)  Other forms of spiritual assessment, like spiritual lifemaps, spiritual ecompas, spiritual ecograms etc., may be worth mentioning (see Hodge, 2015).

(9) The Authors use different abbreviations for religious/spiritual care, e.g., R/SC and S/RS. This should be unified across the manuscript.  

(10)  The article needs thorough proofreading; there are many punctuation mistakes in it.

References

Cornish, M. A., Wade, N. G., & Post, B. C. (2012). Attending to religion and spirituality in group counseling: Counselors’ perceptions and practices. Group Dynamics: Theory, Research, and Practice,16(2), 122–137. https://psycnet.apa.org/doi/10.1037/a0026663

Hathaway, W. L., Scott, S. Y., & Garver, S. A. (2004). Assessing religious/spiritual functioning: A neglected domain in clinical practice? Professional Psychology: Research and Practice,35(1), 97–104. https://doi.org/10.1037/0735-7028.35.1.97

Hodge, David R. (2015). Spiritual assessment in social work and mental health practice. New York: Columbia University Press.

Rosmarin, D. H., Green, D., Pirutinsky, S., & McKay, D. (2013). Attitudes toward spirituality/religion among members of the Association for Behavioral and Cognitive Therapies. Professional Psychology: Research and Practice,44(6), 424–433. https://doi.org/10.1037/a0035218

Vieten, C., Scammell, S., Pierce, A., Pilato, R., Ammondson, I., Pargament, K. I., Lukoff, D. (2016). Competencies for psychologists in the domains of religion and spirituality. Spirituality in Clinical Practice,3(2), 92–114. https://doi.org/10.1037/scp0000078

Author Response

Reviewer 1

Dear Reviewer,

Thank you for reading my paper and for your thoughtful comments and suggestions, which I greatly appreciate.

(1) I have accepted your suggestion for the title of the paper “The Nature of Religious and Spiritual Needs in Palliative Care Patients, Carers, and Families and How They Can be Addressed from a Specialist Spiritual Care Perspective.”

(2) I have added religious coping references in lines 536 – 541. There are references to spiritual struggles (the term that Pargament and Exline use in preference to religious coping) 430-1; 471-2; 490-2; 755-7; 825; 848-9;

(3)  This paper is written from the perspective of a specialist religious, and spiritual care practitioner who has practised for many years in major referral hospitals in Sydney, Australia. I have moved this information to the introduction of the paper for context. 23 -46.

(4) Similarities and differences between spirituality and religiosity added 71-99.

(5) In this hospital, all the staff religious and spiritual care practitioners do work with the therapists at some level the electronic records are the primary way information is communicated. 442-4 added.

(6)   I have suggested the therapist use implicit methods of introduction to religious/spiritual topics. Thank you for the suggested articles which were all helpful and I found Hodge particularly clear.  450 – 457.

7 Thank you for the Vieten et al. reference I used a 2013 edition of this article (Academia) the competencies for psychologists in the domains of religion and spirituality as the hospital library is closed for the Christmas break.  488-9.

(8)  Other forms of spiritual assessment, like spiritual life maps, spiritual ecograms and genograms have been added (Hodge, 2015).

(9)  R/S; R/SC; R/SCP are now consistent across the manuscript

(10)  Apologies for the punctuation mistakes, I have dyslexia and the paper had been sent to an academic editor. Ms Wu at MDPI advised me not to have the paper edited again before returning it to reviewers as it will be edited finally before publication.

Thank you, again for your time and consideration of this paper.

Reviewer 2 Report

1. Please correct multiple instances of anthropomorphizing throughout . ex. 11 - disciplines don't describe, people within disciplines describe... 17 - "this paper follows" - papers don't follow, authors of the paper follow... 51 - "palliative care increasingly clarified itself", palliative care can't clarify, those within the discipline of palliative care have clarified.. 77 "paper will address", again, papers don't address, authors do.  

2. Another general recommendation: go through manuscript and make sure your use of tenses is consistent.

3. "To Examine" in the title is awkward and sets up an incomplete sentence. The use of "Should" in the title sounds a bit preachy.

4. Abstract lacks context for authority of statements. The reader does not know from abstract whether this is a report of research findings, a literature review, an opinion piece, or other - or what methodology was used to compile references upon which the contents of article are based. 

5. Early statistics in the manuscript seem centered in Australia but no account is made of them being generalizable to other areas of the globe.

6. While I find this manuscript very informative and generally well written, I believe the author(s) could assist future readers by constructing a more organized and concise narrative. For example, the roles of religious care and spiritual care providers are addressed in various paragraphs throughout. Perhaps a more abbreviated differentiation early on would suffice. Another example: spiritual assessments are addressed rather early on and then brought up again in a general way in lines 562-3. 

Because it seems as though the author(s) are emphasizing the importance of differentiating the roles of religious care and spiritual care practitioners in palliative care, I'm wondering whether the flow of: a simple of history of religious/spiritual care in hospice/palliative care, differentiating religiosity from spirituality, religious and spiritual care components of palliative care (history taking, care planning, environmental concerns, etc. all addressed in only one place) - whether these can lay the groundwork to describe, compare, and contrast the roles of each type of practitioner and then go on to describing in more detail the roles and needed training of general spiritual care practitioners. All the above points are addressed in this paper, but aspects seem to be scattered throughout, making it a little more difficult for a reader to follow. I think some simple tweaking and streamlining of the narrative would make this a stronger and more readable paper.

7. Has/have the author(s) examined the possibility of spiritual directors/soul companions, who have had much of the training espoused in this article, in the role of general spiritual care provider? See Spiritual Directors International website.

8. The author makes excellent and much needed points, which should further the development of general spiritual care providers in the field of palliative/hospice care. Thank you!

Author Response

Reviewer 2

Thank you for reading my paper and for your thoughtful comments and suggestions, which I greatly appreciate.

  1. Thank you for drawing my attention to anthropomorphising in the paper. I have addressed this matter.
  2. Apologies for the tense mistakes. I have dyslexia and the paper had been sent to an academic editor. Ms Wu at MDPI advised me not to have the paper edited again before returning it to reviewers as it will be edited finally before publication.
  1. New title “The Nature of Religious and Spiritual Needs in Palliative Care Patients, Carers, and Families and How They Can be Addressed from a Specialist Spiritual Care Perspective.”
  2. New abstract lines 5 – 19.
  3. This paper is written from the perspective of a specialist religious, and spiritual care practitioner who has practised for many years in major referral hospitals in Sydney, Australia. I have moved this information to the introduction of the paper for context. 23 -46.
  4. I have attempted to group topics together in a more organised and concise narrative.

A simple history of religious/spiritual care in hospice/palliative care has been added 105 – 120.
Differentiating religiosity from spirituality 72-99.

The section, narrow and broad religious care and spiritual care practitioners, were brought forward in the paper to lines 263 - 300 and abbreviated, later on.

I have removed an early reference to spiritual assessment.

  1. Spiritual directors, non-directive counsellors, and soul companions have been added 469-71.

Thank you, again for your time and consideration of this paper.
